# Impact of the Timing of Spay-Neuter Related to Transport on Disease Rates in Relocated Dogs

**DOI:** 10.3390/ani10040630

**Published:** 2020-04-06

**Authors:** Erin Doyle, Maya Gupta, Miranda Spindel, Emily D. Dolan, Margaret R. Slater, Stephanie Janeczko

**Affiliations:** 1Shelter Medicine Services, American Society for the Prevention of Cruelty to Animals (ASPCA^®^), New York, NY 10128, USA; stephanie.janeczko@aspca.org; 2Strategy and Research, American Society for the Prevention of Cruelty to Animals (ASPCA^®^), New York, NY 10128, USA; maya.gupta@aspca.org (M.G.); emily.dolan@aspca.org (E.D.D.); margaret.slater@aspca.org (M.R.S.); 3Shelter Medicine Help, Fort Collins, CO 85021, USA; miranda.spindeldvm@gmail.com

**Keywords:** animal relocation, animal transport, spay-neuter, surgical sterilization, animal shelter, canine infectious disease

## Abstract

**Simple Summary:**

Companion animal relocation programs transport animals from shelters in areas with a large homeless pet population (source shelters) to shelters in areas in which there is a large demand for adoptable animals (destination shelters). These programs reduce the need for euthanasia caused by shelter overcrowding and resource disparity. However, there is a risk of disease in transported animals due to increased potential for infectious disease exposure and the impact of transport stressors on disease susceptibility. This study evaluated whether the timing of spay-neuter surgery relative to transport impacted the rate of disease in relocated dogs. We found no increase in disease rates in dogs spayed or neutered within two weeks of arrival at the destination shelter as compared to dogs spayed or neutered within the two weeks prior to transport at the source shelter.

**Abstract:**

Companion animal relocation programs are an important method to address geographic and resource disparities in pet overpopulation through transport from areas with high homeless pet populations to areas with high adopter demand. Despite mitigation by following best practices, a potential risk of animal relocation is increased disease incidence related to infectious disease spread and the effects of stress during transport. Surgical sterilization may compound disease risk due to the impact of surgical stress on disease susceptibility and the potential for disease exposure from other patients. Our study aimed to provide information about disease and surgical complication incidence as relates to the timing of surgical sterilization in relocated dogs. A population of 431 dogs relocated to a shelter in Washington State was monitored for disease while at the destination shelter and immediately post-adoption. No increased disease incidence was identified for dogs altered within two weeks of transport at the destination shelter compared with those altered within two weeks prior to transport at the source shelter. Because of disparities addressed by relocation programs, surgical sterilization of relocated companion animals is typically best performed at the destination shelter. Our study indicates that disease incidence is not increased by spay-neuter at the destination shelter.

## 1. Introduction

Approximately 3.3 million dogs enter animal shelters in the United States every year [1]. While euthanasia remains the outcome for approximately 670,000 of these dogs [1], the overall percentage of shelter dogs euthanized has significantly decreased since 2011 [1]. The rate of decline in euthanasia, however, has not been consistent across the country. According to one report, while 1.36 dogs per thousand people were euthanized in 2013 in Colorado, nearly five times that number of dogs were euthanized per thousand people in North Carolina [2]. 

Companion animal relocation programs strive to address this disparity through transport of animals from areas of high homeless pet population to areas with high adopter demand. Successful programs reduce euthanasia and overcrowding at source shelters, allowing these typically under-resourced shelters to focus on initiatives to improve animal welfare and to reduce companion animal overpopulation in their communities.

However, animal relocation programs have the potential for negative consequences including the spread of infectious disease. This disease risk was first documented in the literature through research demonstrating a high prevalence of infectious disease in animals relocated following Hurricane Katrina [3]. While the risks associated with animal movement following natural disasters are not fully analogous to those associated with planned relocation programs, research following Hurricane Katrina contributed to the development of companion animal relocation best practices. 

These best practices have been outlined in several animal relocation guideline documents and were designed in part to mitigate the risk of infectious disease spread and to aid in the formation of responsible, well-planned relocation efforts [4,5,6]. Nonetheless, scientific data evaluating these best practices remain limited. Recent articles exploring transport decision-making [7] and infectious disease in transported felines [8] have begun efforts to objectively analyze non-disaster-related relocation efforts.

One important area of transport logistics that lacks robust scientific data is the scheduling of surgical sterilization. The Association of Shelter Veterinarians’ Guidelines for Standards of Care in Animal Shelters states that animal shelters should require that all adopted cats and dogs be spayed or neutered [6]. The most effective way to accomplish this is through spay and neuter prior to adoption, but research is limited to support recommendations regarding the timing of sterilization relative to relocation. The recent American Veterinary Medical Association and Association of Shelter Veterinarians’ Best Practices for Relocation of Dogs and Cats for Adoption Within the United States [4] and the Companion Animal Transport Programs Best Practices from the Association for Animal Welfare Advancement [5] do not specify sterilization location other than that it should be mutually determined by relocation partners and that waiting for sterilization should not be a barrier to transport.

Ultimately, transport partners must reach an agreement regarding the timing and location of surgical sterilization. In addition to best practice guidelines, numerous factors may influence this agreement including the relative resources of each shelter. Given the impetus for relocation programs, destination shelters are generally better resourced and more likely to have surgical capacity for spay-neuter surgery relative to the source shelter. Performing surgical sterilization at the destination shelter also allows the source shelter to focus their spay-neuter resources on local efforts to reduce pet overpopulation, ultimately contributing to a long-term solution to the resource disparity.

Surgical capacity in the context of shelter length of stay should also be considered. Increased length of stay in the shelter increases the risk of disease exposure and the impact of shelter stressors on disease susceptibility. Prolonged shelter lengths of stay on a population level also contribute to shelter overcrowding. The negative aspects of long lengths of stay are typically more acute in the source shelter given the impact of pet overpopulation on infectious disease risk and shelter overcrowding. Given best practice recommendations to wait at least 48 h after surgical sterilization prior to transport [5], spay-neuter surgery at the source will prolong length of stay at the source shelter by several days if not longer. Ultimately, surgical timing should be planned to minimize shelter length of stay at both partner shelters but with an emphasis on reduction in stay at the source shelter.

While resource and length of stay considerations generally favor sterilization at the destination shelter, we are not aware of prior published data to elucidate whether sterilization at the destination as compared to the source impacts disease rates in relocated animals. The prevalence of respiratory pathogens in shelter dogs has been shown to be high even when clinical signs are absent [9,10]. Surgery and transport may precipitate the development of clinical signs through the impact of stressors on disease susceptibility and/or exposure of uninfected dogs. An increased risk of disease as a sequela of surgery followed by transport or vice versa would significantly impact best practice recommendations for spay-neuter timing in relocation programs.

This study was designed to describe the potential differences in certain diseases in relocated dogs who were altered at source shelters compared to dogs altered at a destination shelter. These data were collected in 2016 from a small sample of relocated dogs at a shelter in southwest Washington. The goal of this descriptive study was to direct programs as well as to begin to fill in the large data gaps in animal welfare regarding relocation programs. This descriptive study aimed to demonstrate whether the timing of sterilization related to transport impacted the frequency of common diseases and/or surgical complications.

## 2. Materials and Methods

### 2.1. Study Site and Population

This study was performed at a private nonprofit shelter with municipal contracts for shelter services in the Pacific Northwest. Data were collected from May to August of 2016. The shelter took in a total of 3531 dogs in 2016, excluding dogs brought in for owner-intended euthanasia. Dogs transferred into the shelter from local and out-of-state partner organizations accounted for 41.4% of canine intake. The remaining dogs brought into the shelter in 2016 were primarily taken in as strays (39.9%) or were relinquished by their owner (17.6%). Adoption was the most common outcome for dogs in this shelter, with 2580 dogs adopted in 2016. As expected, given the shelter’s municipal contracts, return-to-owner was another significant outcome type with 771 dogs reunited with their owners in 2016.

Data points were collected on all 691 dogs transferred into the shelter during the four-month study period in 2016 from all shelter partners, excluding the local shelter coalition. Local transfers are not generally associated with the same disease risks as relocated dogs because of the short duration of transport and the commonality of endemic infectious disease in the communities. Data analysis was then restricted to dogs spayed or neutered within the two weeks preceding transport or the two weeks after arrival at the destination to better elucidate the impact of spay-neuter on disease rates. The two-week window was utilized due to the incubation period of many common shelter-acquired infectious diseases. Beyond two weeks, the relationship between surgical sterilization, disease and transport was presumed to be minimal and a two-week delay in performing spay-neuter (barring special circumstances) would be well beyond the recommended timing of sterilization suggested by best practices to minimize shelter length of stay.

The shelter continued to use its own preexisting criteria at the time for acceptance of dogs via transport. These criteria included a certificate of veterinary inspection within ten days prior to transport, no evidence of ectoparasites or treatment administered if noted, no significant dental issues, and a negative heartworm antigen test performed within thirty days prior to transport for dogs over six months of age. The source shelters were expected to administer a parenteral modified-live multivalent vaccine protecting against Canine Distemper Virus, Canine Parvovirus, Canine Adenovirus, and Parainfluenza Virus as well as an intranasal Bordetella vaccine to all canines at the time of intake. The shelter required dogs over six months of age to have received these vaccines within the six months prior to relocation and required at least two booster vaccinations spaced apart by two weeks of the parenteral intake vaccine prior to relocation for puppies between eight weeks and six months of age. Updated best practices no longer recommend holding puppies for booster vaccinations in source shelters due to increased risk of disease exposure caused by prolonged length of stay outweighing the benefit of booster vaccination [5].

### 2.2. Data Collection

All data were based on information gathered at the study shelter and animal information provided to the study shelter by the source agencies. Spaying (of bitches) or neutering (of male dogs) for animals sterilized prior to transport was collected by the destination shelter based on physical examination of the animal and review of medical records from the source shelter. For dogs sterilized at the destination shelter, the decision regarding the technique for surgical sterilization were per normal shelter surgical protocol and/or surgeon discretion. A record was made of any incidence of spay-neuter complications as well as infectious diseases that may be common or of particular concern in a shelter setting. These included canine infectious respiratory disease (CIRD), gastrointestinal disease, dermatologic disease, or other infectious disease, including zoonoses such as rabies, in the study animals at any point during their stay at the study shelter, as determined by case definitions developed prior to starting data collection. Affected dogs were initially identified during rounds by a licensed veterinary technician and were included in the study if the severity of concern warranted follow-up with a veterinarian and/or treatment beyond monitoring. Table 1 includes the case definitions used for data collection. Two veterinarians were responsible for shelter animal veterinary care during the study period. Diagnosis of disease subcategories defined as “diagnosed by the shelter veterinarian” were left to individual veterinary clinical judgment.

Follow-up was also performed via a phone call with adopters thirty days post-adoption. A script was provided to facilitate consistent conversations with adopters with the goal of determining if the dog was still in the home and if any of the previously listed disease concerns had been diagnosed by a veterinarian after adoption. This script is included as Appendix A.

### 2.3. Statistical Analysis

Statistical analyses were conducted using Stata/IC 15.1 (StataCorp LP, College Station, TX, USA). Frequencies and percentages were used to describe the sample. Associations between surgical altering performed at the source shelter versus study shelter and disease/surgical complications in the study shelter or at post-adoption follow-up were tested using Pearson’s chi squared test or Fisher’s exact test if an expected cell value was below 5. Multiple logistic regression was run to examine predictors of surgical complications (yes/no). The model included binary independent variables for whether the dog had been altered at the destination, had been transported >50% of the journey by flight, had spent more than 3 h in transport, and had a length of stay (LOS) at the destination of more than 1 week and age comparing puppies (less than six months of age) to adult dogs. Associations where *p* < 0.05 were considered statistically significant. Assumptions for logistic regression were checked.

## 3. Results

### 3.1. Total Population

Of the 691 dogs transported during the study period, 435 were altered within the two weeks preceding or following relocation. Four dogs were dropped from the study because their surgery/disease data were incomplete, resulting in a sample size of 431. Of these 431 dogs, 183 were female (42%) and 248 were male (58%). The mean age of the dogs in this sample was two years old (SD = 2; range = 0.1–8 y). There were 68 puppies in the sample (16%), eight of which were altered at the source shelter. Puppies did not differ from adults in their incidence of surgical complications, CIRD, or gastrointestinal disease. While there was a slight difference in skin disease incidence at follow-up, it was a very small number of dogs (3/46 puppies and 1/236 adults with follow-up disease information available). Therefore, all age dogs were combined for the purposes of analysis.

Most of the dogs (305 (71%)) had been transported by a single organization, while the rest were transported by other relocation groups. Flight was the primary mode of transport (>50% of their journey) for 314/431 dogs (73%), while 117/431 dogs (27%) were transported primarily by ground. The mean distance traveled was 1088 miles (SD = 710; range = 675–7000 m), and the mean duration of transport was 7 h (SD = 7; range = 2–48 h).

### 3.2. Surgical Timing

Surgical alteration was performed at the source for 106 dogs (25%): 13 % neutered and 12% spayed. Surgical alteration within two weeks of arrival at the destination was performed for 325 dogs (75%): 45% neutered and 31% spayed. 

### 3.3. Animal Outcomes

The mean length of stay at the destination shelters was 9 d (SD = 11; range = 1–87 d). The outcome was adoption for 427 (99%) of the dogs, and four dogs (1%) were transferred.

There were 321 dogs (75%) whose adopters were reached at 30-day follow-up contact. Of those, 277 (86%) were still in the home at 30-days. Forty-one animals (10% of the 423 for whom return information was available from the shelter) were returned within 30 d. 

Ultimately, some disease information was available from 282 adopted dogs whose adopters were reached at 30-day follow-up.

### 3.4. Disease Rates

A summary of overall disease incidence by category is provided in Table 2. 

#### 3.4.1. Spay-Neuter Complications in the Shelter

Only nine of the 431 dogs altered and with relevant data recorded had complications reported in the shelter following surgical alteration. Two of the nine complications (22%) were mild, and seven were (78%) moderate.

No significant relationship between surgical timing and spay-neuter complications in the shelter was found. The rate of spay-neuter complications in the shelter among dogs altered within two weeks of transport at the source (2% (2/106)) did not differ from the rate of complications among those altered within two weeks of arrival at the destination (2% (7/325, 95% CI: 1–4%)) *p* = 1.0, Fisher’s exact test).

#### 3.4.2. Spay-Neuter Complications at Follow-up

Of the 282 dogs with post-adoption information on spay-neuter complications, 15 had complications reported at follow-up: five (34%) showed mild complications, and ten (67%) showed moderate complications. None of these dogs had shown any surgical complications in the shelter. No significant relationship between surgical timing and spay-neuter complications at follow-up (*p* = 1.0, Fisher’s exact test) was found within this subgroup.

#### 3.4.3. CIRD in the Shelter

There were 19 (4%) dogs (17 uncomplicated, two complicated, and zero Canine Distemper Virus) noted to have developed CIRD while in the shelter (mean age: 2.6, range = 2 mo–5 y). Five of these dogs were among the 106 altered prior to transport (5%), and 14 were of the 325 (4%) altered after. No significant association was found between surgical timing and CIRD in the shelter (χ^2^(1) = 0.03, *p* = 0.9).

#### 3.4.4. CIRD at Follow-up

Of the 282 dogs with follow-up information on disease rates, 49 animals showed signs of CIRD at follow-up (17%): 24 uncomplicated, 25 complicated, and zero Canine Distemper Virus. Three of the dogs had already been diagnosed with CIRD in the shelter. Twelve of these dogs had been altered prior to transport (12/106, 11%), and 37 were altered after arrival (37/327, 11%). The mean age for this group of dogs was 1.7 y (range = 2 mo–5 y). No significant association was found between surgical timing and respiratory disease at follow-up (χ^2^(1) = 0.4, *p* = 0.6).

#### 3.4.5. Gastrointestinal Disease in the Shelter

No cases of gastrointestinal disease in the shelter were recorded.

#### 3.4.6. Gastrointestinal Disease at Follow-up

Of the 280 animals for whom follow-up information on gastrointestinal disease was available (two dogs had these data missing), five animals (2%) were reported by their adopters to have had gastrointestinal disease post-adoption. One dog was reported by the adopter to have been diagnosed with Canine Parvovirus by a veterinarian. The mean age of these dogs was one year (range = 4 mo–2 y). No significant association was found between surgical timing and gastrointestinal disease at follow-up (χ^2^(1) = 0.01, *p* = 0.9).

#### 3.4.7. Skin Disease in the Shelter

There were five dogs (1%) who showed signs of skin disease in the shelter; none were diagnosed with sarcoptic mange or dermatophytosis, and thus, all five incidents of skin disease were classified as “other”. All five of these dogs had been altered after arrival at the destination shelter. The mean age for this group of dogs was 2.7 y (range = 4.5 mo–7 y). No significant relationship was found between surgical timing and skin disease in the shelter (χ^2^(1) = 1.7, *p* = 0.2). 

#### 3.4.8. Skin Disease at Follow-up

Of the 280 animals for which follow-up information about skin disease was obtained (two dogs had these data missing), four animals were reported to show signs of skin disease at follow-up, one of which was categorized as dermatophytosis and three of which were categorized as “other”. None of the dogs had been diagnosed with skin disease in the shelter. All four of these dogs had been altered after arrival. The mean age of these dogs was 3.5 mo (range = 2–7 mo). No significant association was found between surgical timing and skin disease at follow-up (*p* = 0.6, Fisher’s exact test). 

#### 3.4.9. Other Infectious Disease in the Shelter

No cases of other disease were recorded in the shelter. 

#### 3.4.10. Other Infectious Disease at Follow-up

No cases of other infectious disease were reported at follow-up. 

#### 3.4.11. Multiple Logistic Regression Analysis

A multiple logistic regression model was developed using a combined outcome of any spay-neuter complications and/or disease at any point. Four dogs were missing information about having disease (0.9%), 91 dogs had disease at some point (21%), and 340 dogs had no disease recorded (79%). No difference was found in the odds of dogs altered within two weeks of arrival at destination to develop disease at any point compared to dogs altered within two weeks of transport at source, even when controlling for transport methods, time, LOS at the destination, and age (see Table 3).

## 4. Discussion

This is the first study that we are aware of that explores the impact of the timing of surgical sterilization on disease rates in relocated dogs. Data from this study demonstrated no difference in the incidence of disease in dogs altered at the source shelter versus those altered after arrival at their destination shelter. Factors including surgical capacity, shelter length of stay, and community need typically favor surgical sterilization of relocated animals at destination shelters. This study’s findings coupled with the increased disease risk associated with a longer length of stay at source shelters [11] support surgical sterilization at the destination shelter.

Animal information and overall disease incidence in the study population was generally similar to available information on comparable populations and was in keeping with the authors’ expectations based on the typical requirements and goals of animal relocation programs. Nearly all dogs in the study population were ultimately adopted. This is to be expected given that relocation programs move companion animals from shelters in communities with pet overpopulation to shelters in communities with high adopter demand. Approximately 86% of dogs remained in the adopted home 30 d post-adoption, a proportion that is on par with previously reported data regarding retention in the home post-adoption [12]. 

More dogs in the study were altered at the destination shelter. This is expected given that the destination shelter’s local canine intake was low enough to warrant receiving transported dogs, thus presumably leaving excess surgical capacity to accommodate the sterilization of the transported dogs. The larger sample size of dogs altered within two weeks of arrival at the destination shelter further strengthens the assertion that spay-neuter at the destination does not increase disease rates.

Spay-neuter complication rates were on par with previously published data. Two percent of study dogs experienced a spay-neuter complication during their shelter stay, while five percent were noted to have a spay-neuter complication within thirty days of adoption. These figures are comparable with the complications rates of 5.4–7% [13] and 7.5% [14] reported in two previous studies evaluating spay-neuter complications of canines undergoing orchiectomy and/or ovariohysterectomy surgeries. Ultimately, no significant relationship between spay-neuter timing relative to transport and risk of complication was identified.

Four percent of study dogs were diagnosed with CIRD at the destination shelter, while approximately 17% of study dogs for whom post-adoption information was available were diagnosed with CIRD within the first 30 d of adoption. No published data were identified to determine whether this is on par with average levels of infectious respiratory disease in relocated dogs. Limited published information exists regarding CIRD incidence in shelter dogs regardless of intake source. One study found a lower incidence in shelter dogs than in the study population [11], but CIRD is commonly reported to be of high morbidity in shelter dogs [15]. CIRD levels in the study population is in keeping with the incidence of respiratory disease of relocated animals reported by destination shelters (17.5%) via post-transport reports (PTR) for the ASPCA’s relocation program (ASPCA unpublished data—2018 Q4 PTR). There was no impact of spay-neuter timing on CIRD incidence.

Many factors have the potential to impact disease rates in relocated animals. Multiple logistic regression analysis confirmed a lack of significant difference in disease rates based on spay-neuter timing when controlling for other potential compounding factors including transport method, time, length of stay at the destination, and age. A difference in disease rates as low as five percent could create concern for shelter population health, yet this study demonstrates a difference in disease incidence based on spay-neuter timing that is well below that measure of clinical significance and, in fact, was less than one percent in this population. 

While this study provides compelling initial information about the impact of spay-neuter timing on disease rates in relocated dogs, some limitations are important to consider. A majority (73.1%) of the study dogs were transported via air. A published survey of shelters nationwide indicated that only 34% of relocated dogs were transported via air [7]. Though our multivariate analysis did not show a difference based on transport method in the odds of developing disease, the difference in our study population due to the frequency of air travel may limit generalizability of this sample to the broad population of dogs who are relocated nationally.

Another limitation is the low overall disease levels for gastrointestinal disease, skin disease, and any other infectious diseases with only one case each recorded for Canine Parvovirus and dermatophytosis, both significant pathogens in a shelter setting. As such, little information was gathered about the impact on certain diseases of note, including zoonotic pathogens. As with CIRD, no impact of the timing of surgical sterilization was identified on disease rates for gastrointestinal or skin disease but the low number of cases make the significance of this association uncertain.

Data about disease rates for other dogs beyond the study population at either the source or destination shelters were not obtained. Analysis of disease incidence in other dogs in the source and destination shelters would further isolate the impact of spay-neuter timing. Similarly, the timing of disease was not tracked based on its relation to shelter length of stay. Further evaluation of the impact of shelter length of stay on CIRD incidence and its interplay with spay-neuter timing in relocated dogs is warranted.

Finally, data were not available on dogs who were removed from the transport roster based on disease identified prior to relocation. As such, disease and/or complication rates may be higher in dogs sterilized at the source shelter than is reflected in this data set. Further data about disease rates including dogs removed from the transport roster would better define overall disease incidence related to spay-neuter timing.

This study was limited to a population of relocated dogs at a single destination shelter following best practices at the time. Dogs in this study were not assigned to each group, rather they were tracked in an applied way through the regular course of their relocation. Our findings indicate that there is a relatively low incidence of disease in dogs transported via well-managed relocation programs regardless of whether they are sterilized at the source or the destination and that this would likely apply to similar shelters and locations. While geographic and community-based variations in companion animal disease prevalence could impact disease rates for relocation programs in other parts of the United States [16] or could involve more urban or rural shelter partner communities [17], these variations would not likely result in findings very different from those presented here. Further research exploring disease rates in relocated dogs at different organizations, locations, and populations will be important. One important next step to further elucidate the impact of spay-neuter timing on disease in relocated dogs would be to examine disease rates in an experimentally assigned population of dogs from a source shelter, looking at the impact of surgical timing in consideration of length of stay and disease rates in the source shelter as a whole.

## 5. Conclusions

Spay-neuter timing is one of many important decisions to be considered in structuring an effective relocation program. Our findings indicate no increase in disease incidence when sterilization is performed within two weeks after transport at the destination shelter compared to within two weeks prior to transport at the source shelter. These study findings begin to provide evidence to alleviate disease-related concerns regarding spay-neuter timing, allowing the decision to focus on other factors such as relative resources, shelter length of stay, and community needs. These factors typically favor surgery at the destination shelter. This study provides an early and important piece of evidence in a growing body of research to guide companion animal relocation best practices.

## Figures and Tables

**Table 1 animals-10-00630-t001:** Definitions utilized to identify and categorize disease identified in relocated dogs.

Disease Category	Sub-Category	Definition
Spay-neuter complications	mild	Redness, itching, licking, or mild seroma present but no medication was required
	moderate	Surgical complications requiring pain medication and/or antibiotics
	severe	Anesthesia required to repair surgical complications such as a dehiscence or hematoma
Canine Infectious Respiratory Disease (CIRD)	uncomplicated	Respiratory signs such as nasal discharge, coughing, and/or sneezing that warranted a veterinary examination but where only seven days of medication or less was required
	complicated	Clinical signs as described above that warranted a veterinary examination and treatment beyond seven days of medication
	Canine Distemper Virus	As diagnosed by the shelter veterinarian
Gastrointestinal disease	Canine Parvovirus	As diagnosed by the shelter veterinarian
	other	Clinical signs such as vomiting and diarrhea that warranted a veterinary examination but were not diagnosed as caused by Canine Parvovirus
Dermatologic disease	Sarcoptic mange (canine scabies)	As diagnosed by the shelter veterinarian
	Dermatophytosis	As diagnosed by the shelter veterinarian
	other	fur loss or other dermatologic signs that warranted a veterinary examination but were not diagnosed as caused by sarcoptic mange or dermatophytosis

Other information gathered about each study animal included age, gender, intake date to the source shelter, intake date to the study shelter, location (source or study shelter) and date of spay-neuter surgery if known, and the name and location of the source agency. Information collected about the relocation process included the agency responsible for relocation if applicable, transport type (>50% flight and >50% ground/driven), total time transported if known, and distance transported if known.

**Table 2 animals-10-00630-t002:** Disease incidence by category identified while at the destination shelter or at post-adoption follow-up.

Disease	In the Shelter, *n* = 431			At Follow-up, *n* = 282		
	Frequency	Percent	95% CI	Frequency	Percent	95% CI
Spay/neuter complications	9	2%	1–4%	15	5%	3–9%
CIRD	19	4%	3–7%	49	17%	13–22%
Gastrointestinal	0	0%	na	5	2%	0.6–4%
Skin	5	1%	0.4–3%	4	1%	0.4–4%
Other	0	0%	na	0	0%	na

CI = confidence interval, na = not applicable.

**Table 3 animals-10-00630-t003:** This logistical regression model shows the odds ratios (OR) of any disease by location of altering, controlling for other factors.

*n* = 431	Frequency	OR (95% CI)	*p*-Value
Any Disease Yes	Any Disease No
Altered at destination
Yes	69	256	1.0 (0.6–1.7)	1.0
No	22	84	ref.	
Transported by air
Yes	63	251	1.2 (0.4–3.3)	0.8
No	28	89	ref.	
Transport time > 3 h
Yes	32	98	1.6 (0.6–4.3)	0.4
No	59	242	ref.	
LOS > 1 w
Yes	45	141	1.4 (0.9–2.3)	0.2
No	46	199	ref.	
Puppy
Yes	45	171	1.0 (0.6–1.6)	1.0
No	46	169	ref.

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
