# Peer review of "Impact of the Timing of Spay-Neuter Related to Transport on Disease Rates in Relocated Dogs"

_animals, 2020, doi:10.3390/ani10040630_

Round 1
Reviewer 1 Report
The authors have done an excellent job in presenting the need for this type of research, and have introduced how to perform a descriptive study to start supporting evidence-based guidelines for transporting shelter animals. Because the manuscript was written so well, I only have a few comments and questions for the authors/editors to consider.
1) Line 99: There is mention that there is no published data on whether sterilization impacts disease rates in relocated animals. Consider looking for/mentioning prior published data on peri-operative complications surrounding shelter animal transport.
2) Data Collection section starting at Line 151: There was no mention of looking for zoonotic pathogens or pathogens of significance for public health. This may be worth mentioning somewhere in the article. Consider explaining why this was not included in the study.
3) Lines 287-88: Authors should consider giving a citation/evidence from the literature to back up the statement "increased disease risk associated with a longer length of stay at source shelters...". If there is not sufficiency evidence, consider re-wording.
4) Line 350-51: I was hoping to see a little more discussion on if the authors felt these findings could be generalizable to other regions of the country. And why/why not? i.e. regional disease differences between the southeastern states and west coast, etc.
5) Finally, in the discussion, it would be awesome for future researches if the authors would discuss how to move beyond a descriptive study and what type of future research would provide significant impact to shelter animal transport.
Again, overall, this is an excellent paper, well-written, and well thought out. It was a pleasure reviewing this manuscript.
Author Response
Dear reviewers,
Thank you for your kind words and helpful feedback on our manuscript. Please see the comments below where we outline the edits made in response to your comments. Also, please note we updated the section on AVMA and ASV guidelines on lines 69-79 to reflect recent changes to their recommendations.
Reviewer 1
The authors have done an excellent job in presenting the need for this type of research, and have introduced how to perform a descriptive study to start supporting evidence-based guidelines for transporting shelter animals. Because the manuscript was written so well, I only have a few comments and questions for the authors/editors to consider.
We appreciate your positive feedback on our manuscript.
1) Line 99: There is mention that there is no published data on whether sterilization impacts disease rates in relocated animals. Consider looking for/mentioning prior published data on peri-operative complications surrounding shelter animal transport.
In addition to searching for literature on sterilization and disease impacts on relocated animals we have also searched thoroughly for cited literature for ASV spay-neuter guidelines, cited references for the AVMA relocation best practices, peri-operative complications shelter animal transport, transport animal sterilization, transport animal spay, transport animal complications, transport animal surgery, relocation spay neuter, relocation animal sterilization, canine transport complications, canine relocation, and canine transport spay neuter. We acknowledge that there may be literature we have not seen and have changed the language on line 100 to reflect not that there isn’t any literature but that we are unaware of any.
2) Data Collection section starting at Line 151: There was no mention of looking for zoonotic pathogens or pathogens of significance for public health. This may be worth mentioning somewhere in the article. Consider explaining why this was not included in the study.
We agree that mentioning zoonotic pathogens is worth discussing. We did include ringworm in the study specifically because it is a prominent zoonotic pathogen. Other zoonotic pathogens like rabies would have been detected in the “other” category that we originally planned to look at but we did not have any cases in the data. We made a comment on lines 161-162 of the paper indicting that we would have detected rabies and other zoonotic illness if we had found any cases. We also edited lines 345-348 to indicate our low zoonotic infectious disease rates in the limitations.
3) Lines 287-88: Authors should consider giving a citation/evidence from the literature to back up the statement "increased disease risk associated with a longer length of stay at source shelters...". If there is not sufficiency evidence, consider re-wording.
Thank you for pointing out this place that needed literature to support it. We have added a citation for Edinboro, 2004.
4) Line 350-51: I was hoping to see a little more discussion on if the authors felt these findings could be generalizable to other regions of the country. And why/why not? i.e. regional disease differences between the southeastern states and west coast, etc.
This is an important point. We do think that there could be regional disease differences, especially, as the reviewer mentions, in the southeastern states. However, we feel these differences are not so significant as to preclude generalizability. We have added information in the limitations (see lines 364-371) and included citations from Banfield Pet Hospital 2014 report and DiGangi 2019 describing geographic and community-based disease prevalence variations.
5) Finally, in the discussion, it would be awesome for future researches if the authors would discuss how to move beyond a descriptive study and what type of future research would provide significant impact to shelter animal transport.
Thank you for suggesting this. We added what we think would be the most important next steps for this research on lines 372-375.
Again, overall, this is an excellent paper, well-written, and well thought out. It was a pleasure reviewing this manuscript.

Reviewer 2 Report
In this paper the moment of spaying a shelter dog in relation to transport, in term of disease rates, is examined. Likewise these paper provides important finding for future recommendations regarding the scheduling of surgical castration in relation to relocation.
Some small remarks:
- Is it possible to give us some insights in the ratio ovariectomy vs ovariohysterectomy in these cases? Did you observe a diffence in the incidence of the diseases when comparing these 2 surgical approaches?
- Line 191: the average age of the dogs is 2 years: is this the average or the median? What is the youngest dog what is the oldest?
- Line 203: Can you explain what is the difference in your definition between spay and neuter?
- Line 209-210: I’m a bit lost here in the numbers: what is the difference between 321 dogs, the 423 dogs and the 282 dogs (in table 2). Please clarify.
- Line 220: what do you mean with surgical status?
- Line 230: is there any effect of ‘timing of surgery’ ?
Author Response
Reviewer 2
In this paper the moment of spaying a shelter dog in relation to transport, in term of disease rates, is examined. Likewise these paper provides important finding for future recommendations regarding the scheduling of surgical castration in relation to relocation.
Some small remarks:
- Is it possible to give us some insights in the ratio ovariectomy vs ovariohysterectomy in these cases? Did you observe a diffence in the incidence of the diseases when comparing these 2 surgical approaches?
Unfortunately, we did not collect data about whether the surgery performed was ovariectomy or ovariohysterectomy. We do think this would be interesting information to collect in a future study.
- Line 191: the average age of the dogs is 2 years: is this the average or the median? What is the youngest dog what is the oldest?
Two years of age is the average age of dogs in this study. The median age is 1.5 years and the dogs ranged from 0.1-8 years of age. This data was included in the paper on line 199.
- Line 203: Can you explain what is the difference in your definition between spay and neuter?
We have added a more complete description of how spay/neuter data was collected from the source and how it was executed at the destination shelter on lines 154-158.
- Line 209-210: I’m a bit lost here in the numbers: what is the difference between 321 dogs, the 423 dogs and the 282 dogs (in table 2). Please clarify.
We added language on lines 261-220 to clarify that we reached 321 dogs’ adopters for follow-up and we had disease information on 282 of those dogs. The 423 number is the subset of animals for whom we knew if they were returned or not.
- Line 220: what do you mean with surgical status?
Surgical status referenced the timing of spay-neuter relative to transport. We have changed the term to surgical timing to better clarify.
- Line 230: is there any effect of ‘timing of surgery’?
We found no effect of the timing of surgery and we changed the language throughout from surgical status to surgical timing.